# The shared experiences of insured members and the uninsured in health care access and utilization under Ghana's national health insurance scheme: Evidence from the Hohoe Municipality

Suraiya Umar[1], Adam Fusheini[2,3]*, Martin Amogre Ayanore[4]

1 Department of Population and Behavioural Sciences, School of Public Health, University of Health and Allied Sciences, Ho, Ghana, 2 Department of Preventive and Social Medicine, Otago Medical School, University of Otago, Dunedin, New Zealand, 3 Center for Health Literacy and Rural Health Promotion, Accra, Ghana, 4 Department of Health Policy, Planning and Management, School of Public Health, University of Health and Allied Sciences, Ho, Ghana

* adam.fusheini@otago.ac.nz

**Data Availability Statement:** All relevant data are within the manuscript and its Supporting information files.

## Abstract

### Background

The National Health Insurance Scheme (NHIS) was introduced in Ghana in 2003 to remove financial barriers and to promote equitable access to health care services. Post implementation has been characterized by increases in access and utilization of services among the insured. The uninsured have been less likely to utilize services due to unaffordability of health care costs. In this study, we explored the experiences of the insured members of the NHIS, the uninsured and health professionals in accessing and utilizing health care services under the NHIS in the Hohoe Municipality of Ghana.

### Methods

Qualitative in-depth interviews were held with twenty-five NHIS insured, twenty-five uninsured, and five health care professionals, who were randomly sampled from the Hohoe Municipality to collect data for this study. Data was analyzed using thematic analysis.

### Results

Participants identified both enablers or motivating factors and barriers to health care services of the insured and uninsured. The major factors motivating members to access and use health care services were illness severity and symptom persistence. On the other hand, barriers identified included perceived poor service quality and lack of health insurance among the insured and uninsured respectively. Other barriers participants identified included financial constraints, poor attitudes of service providers, and prolonged waiting time. However, the level of care received were reportedly about the same among the insured and uninsured with access to

**Funding:** The author(s) received no specific funding for this work.

**Competing interests:** The authors have declared that no competing interests exist.

quality health care much dependent on ability to pay, which favors the rich and thereby creating inequity in accessing the needed quality care services.

## Conclusion

The implication of the financial barriers to health care access identified is that the poor and uninsured still suffer from health care access challenges, which questions the efficiency and core goal of the NHIS in removing financial barrier to health care access. This has the potential of undermining Ghana's ability to meet the Sustainable Development Goal 3.8 of universal health coverage by the year 2030.

## Introduction

Health insurance systems such as Social Health Insurance (SHI) are considered an essential vehicle to achieving Universal Health Coverage (UHC), and a feasible alternative funding mechanism for the health sector in many low and middle-income countries (LMIC) [1]. SHI is both a mechanism that helps mobilize resources for health, pool risk, and provides more access to health care services for the poor [2] and also seen in developing countries as a pro-poor health care intervention which helps to bridge the financial gap between the rich and the poor in accessing quality health care [3]. Cost is often a barrier to accessing healthcare in times of need for most people, especially the poor [4]. In South Africa, for instance, the first edition of its Future Health Index (FHI), a study which explored how countries address long-term global health challenges through health system integration and technology revealed that cost of healthcare is a major barrier to health care access [5]. Thus, over the past few years, several countries including Ghana have introduced and implemented national health insurance schemes (NHISs) as part of health sector reforms to increase access to and utilization of health services [1].

While some developed countries have successfully ensured optimum levels of healthcare for their entire populace, the case in most developing countries is different. Health care accessibility and utilization remains limited as a result of financial and socio-cultural challenges that face such economies [6]. It is in the light of these that we explored the shared experiences of insured members and the uninsured in health care access and utilization under Ghana's National Health Insurance Scheme.

## National health insurance scheme in Ghana

Ghana's National Health Insurance Scheme is a health policy reform which allows Ghanaian residents to have access to health care services at any time without paying at the point of service [7]. The NHIS was introduced to address the problem of inequity by removing the financial barrier to health care access [8]. The cost barrier or fee-for-service system otherwise referred to as "cash and carry" lasted from the early 1990s under the World Bank and International Monetary Fund (IMF) Structural Adjustment Program (SAP) until 2004 when the current NHIS was implemented in Ghana [9]. The cash and carry system required those seeking care to make upfront payment before health service providers attended to them [10]. The government's efforts at achieving UHC via the NHIS implies that all public health facilities in the country are automatically accredited under the scheme. However, private health facilities need to apply to the National Health Insurance Authority (NHIA) for accreditation before they can

render services under the scheme. Access to and utilization of services under the scheme are by subscription.

Subscription is by payment of an annual premium by those in the informal sector and those who do not contribute to the Social Security and National Insurance Trust (SSNIT) pension scheme. The annual premium varies between twenty-five Ghana Cedis (GHS25) (approximately US$4.30) and GHS46 (US$7.91) between the low- and upper- income groups. Those who contribute to SSNIT only need to pay a registration fee of about GHS5 (US$0.86). There are exemptions for SSNIT pensioners, children less than eighteen years, pregnant women and those seeking postnatal care, the aged above seventy years, the indigent or extreme poor, estimated in 2016 to be about 8.2 percent of Ghana's population [11] and those identified by the Minister for Employment and Social Welfare as needing support [12]. The indigent as defined by Act 650 (2003), which has been repealed by Act 852 of 2012 refers to "a person who does not have a visible or adequate means of income or who does not have a person to support him or her and by the means test qualifies as an indigent" [13].

Access to health insurance is assumed to improve access to healthcare and grant financial protection to the insured [14] as the insured are more likely to seek health care services when they are sick [15]. Yet population coverage by health insurance is reportedly still low, about 38–40% of the population, including even among the poor with a heavily subsidized premium and exemptions provided [8]. Studies have shown that cost of premiums, having large household size, long queues and waiting time, perceived poor quality of drugs, and negative attitude of service providers at both the healthcare facilities and the health insurance offices and being in the lower socioeconomic quintiles are factors associated with refusal to enroll on the NHIS [14,16]. Other reasons include rare episodes of illness, limited benefits of the scheme, access to free drugs, and social security against unforeseen health challenges and poor service quality [16]. However, encouragement by friends, family members, and colleagues were found to play a significant role [17].

Insured members of the NHIS are more likely to seek health care services when they are sick [15], as it serves as a means of using health care services for the insured [18]. In other words, it is an effective option for increased access to health care delivery with increased attendance to health facilities [19]. Again, evidence shows better access to quality formal health care for the insured than the uninsured [20] with the former and their dependents under 18 years having access to free health care services at public and accredited private health facilities while the uninsured have to pay for services when seeking health care. Subscription to the NHIS also provides easy access to specific population groups such as subscribed pregnant women who are said to make more frequent visits to antenatal service facilities, seek and have antenatal care, skilled delivery, and postnatal care [21] compared to women who are non-subscribers [22]. Despite these benefits, it is estimated that the majority of Ghanaians (62%) are not subscribed to the scheme [23]. There have rather been reports of stagnating active membership, poor quality health care rendered to subscribers and increasing costs which have raised concerns on the operational and financial sustainability of the scheme [1,24]. Evidence of decreasing utilization of health care by subscribers due to low staffing and inadequate health facilities under the NHIS have been reported [20].

The uninsured population access to and utilization of health services have also been documented in the literature. The uninsured are less likely to have a usual source of care or a recent visit to a health facility [25] due to having to pay to access health care [20]. Therefore, the uninsured are more likely to choose informal sources of care compared to the insured [26], and are less likely to have access to and utilize health care services.

There are also some socio-economic gradients such as education, employment, insurance status, income and health state that influence utilization of health services in Ghana [27]. This

implies that people are likely to have different and diverse experiences under the same health system depending on a number of factors including ability to pay for and utilize health services. The satisfaction levels and self-rating of services will equally differ as a result of the different experiences. To ensure at least a more equitable experience, most developing countries including Ghana have tried diverse health financing mechanisms including health insurance, to ensure universal access to quality basic health care [24].

Therefore, the attitude of the insured and the uninsured under the NHIS towards health care utilization could be as a result of their experiences while accessing health care. Particularly, the uninsured face a major barrier of paying very high fees for accessing health care services [20]. For the insured, it is common to find that many of them are unwilling to access health care with their NHIS cards due to delays associated with health care utilization with the cards [20]. Others would also prefer to attend private health facilities where they even pay higher fees out-of-pocket to attain quality health care services [20]. Despite the existence of these barriers, there is paucity of qualitative empirical evidence on the experiences of both the insured and uninsured in health care accessibility and utilization using the NHIS as a proxy. The purpose of this study was to compare the experiences of NHIS insured and uninsured in accessing and utilizing healthcare services. Specifically, the study aimed at exploring factors motivating accessibility and utilization of health care services among NHIS subscribers and non-subscribers in the Hohoe Municipality; assessing barriers to accessibility and utilization of health care services by NHIS subscribers and non-subscribers. The study also sought to ascertain possible disparities in health care services provided to NHIS subscribers and non-subscribers; and to assess the views of service providers on the experiences of NHIS subscribers and non-subscribers in accessing and utilizing health care services.

## Conceptual framework: Andersen healthcare service utilization model

In order to explore the experiences of both the insured and uninsured members under the NHIS in Ghana, we employed the Health Care Service Utilization Model developed by Anderson [28] as a guide. The Health Care Service Utilization Model was first propounded by Andersen in 1972 and has since gone through subsequent reviews by Andersen and Newman [29]. This model was developed to help discover the conditions that motivate or impede an individual from utilizing health care services [30]. The model focuses on and describes the role and three characteristics that are considered to influence an individual's use of health care services [30]. These characteristics are pre-disposing, enabling, and need factors. The predisposing factors are mainly socio-cultural characteristics of individuals before experiencing ill-health. These characteristics can be categorized into (i) social structure such as education, occupation, ethnicity, social networks, social interactions, and culture, (ii) health beliefs, such as attitudes, values, knowledge and perceptions of people concerning health care services, or (iii) demographic factors such as gender and age [28].

The enabling factors are described as the external factors that are important in influencing an individual's decision to use health care services [30]. Enabling factors include the presence of a health care facility, the distance between an individual's home to this facility, the cost involved in accessing this facility, health insurance status, and how effective this facility is in addressing health needs. Andersen considered the need factors as the most immediate cause of health service utilization. Perceived need of the individual including health needs or illness severity determines the need for utilizing health care services. Thus, a person will use health care services based on their perceptions of how serious their condition is. In addition, the

professional judgement about people's health status, described as evaluated needs by Andersen can also influence their decision to use health care services [30].

This model was considered relevant to this study due to its strength in revealing need, predisposing and enabling factors that influence health care access and utilization among the insured and uninsured and stakeholders in the health care delivery process as noted by Kuuire et al, [31].

## Materials and methods

### Study design

To examine the experiences of insured and uninsured clients under the NHIS on health care access and use, the study applied a qualitative phenomenological design. This allowed for description of the meaning for several individuals of their lived experience of a phenomenon-the lived experiences of both the insured and uninsured clients about access to and utilization of health services as described by them [32,33]. In other words, the study design provided the opportunity to synthesize common experiences on the subject under investigation [34].

### Study setting

The Hohoe Municipality is located in the central part of the Volta Region of Ghana. It is one of the eighteen districts/municipalities in the Volta Region. The municipality is bounded to the north by the Jasikan District, to the north-west by the Biakoye District, both in the Oti Region of Ghana. The Hohoe Municipality is bounded to the west and south-west by the Kpando Municipality, to the south by the Afadjato South District, all in the Volta Region. The Ghana Statistical Service projected population estimates for the Hohoe Municipality in 2019 was 206,089 [35]. About 55% of the population are engaged in farming. Twenty-five percent are also involved in trading, 15% in livestock rearing and 5% engage in other activities [36]. The Municipality is divided into seven administrative sub-municipalities namely: Akpafu/Santrokofi, Alavanyo, Agumatsa, Lolobi, Gbi-rural, Hohoe-sub and Likpe. It has one hospital with onchocerciasis research center; a research center for malaria; a private community clinic; thirteen government health centers, six Community-Based Health Planning and Services (CHPS) compounds, twenty-six newly functional CHPS zones and a Reproductive and Child Health (RCH) with adolescent health center, all totaling forty-nine health facilities [37]. The Hohoe sub-municipality was purposively selected from among the seven divisions of the Hohoe Municipality for this study due to convenience and proximity to the researcher.

### Recruitment

The Hohoe sub-municipality has twenty-five communities. Out of these, five communities were purposively selected based on convenience and proximity to recruit study participants. The selection of the five communities was also informed by the cosmopolitan nature of the municipality with different ethnic groups, socio-economic activities and demographic characteristics. This was, therefore, to give a cross-sectional representation of the wider Hohoe communities. The five communities selected included; Ahado Central, Zongo East, Abansi, Gbohome, and Kitikpa. They were selected from the north, east, west, south and central communities of the Hohoe sub-municipality. The population of interest in this study was persons insured under the NHIS and those uninsured. Persons who had a valid NHIS membership card and used it actively for accessing health care were classified as insured. All other persons who obtained health care through other payment options outside the NHIS were considered

uninsured. Recruitment of participants was by moving from house to house personally by the first author and talking to household members to identify insured and uninsured.

## Data collection

Data was collected across all the five study communities in the Hohoe sub-municipality. The five communities were purposively selected and at each community level, ten houses were randomly selected from which either one eligible insured or uninsured respondent was sought to participate. Thus, at each household level, an eligible respondent was insured or uninsured but not both. Simple random sampling was used to select respondents at each household level. The process continued across each of the ten houses until five insured and five uninsured respondents were obtained at each community level. In total, twenty-five insured and twenty-five uninsured participants took part in the study across the five communities. Eligible respondents who consented to participate in the in-depth interviews (n = 50) were interviewed in convenient places at home or at work if they preferred so. With open-ended questions in an in-depth interview guide, respondents' views on the experiences relating to need, predisposing and enabling factors for accessing and using health care were elicited. The questions were on the socio-demographic characteristics of study participants, the motivating factors and or the extent to which the health insurance made a difference to subscribers' access to and utilization of health care, the challenges uninsured have in accessing and utilizing healthcare, and health care providers' experiences of providing care to both insured and uninsured under the NHIS among others. Other topics included reasons for not subscribing to the NHIS, non-renewal of health insurance, recommendations for improving the NHIS, barriers to health care utilization and access. Follow-up questions (probes) were posed where necessary to obtain additional insight. Two trained research assistants with knowledge of the study setting and language assisted in audio recording and language translation during interviews. Hand written notes were taken together with audio recordings. The duration of interviews and the number of questions varied from one participant to the other and conducted in Ewe or English depending on a participant's choice. The interviews lasted between fifteen and twenty-five minutes.

Prior to interviews, an information sheet which contained the details of the study was given to potential respondents to read and decide whether or not to be part of the study. After each interview, participants were verbally appreciated for their time, effort, and for agreeing to be part of the study.

## Data analysis

The Braun and Clarke's method of thematic analysis, which involves six phases: becoming familiar with the data; generating initial codes; searching for themes; reviewing themes; defining and naming themes and finally producing the report [38] was employed in the study. Using this thematic analysis process allowed the voices of participants to be at the forefront of analysis [38]. All the recorded interviews were transcribed by the first author. Five of the interviews not in English were translated and then transcribed into English with the assistance of two research assistants, who were natives of the town. Editing of grammar but not the content was done. The transcripts were then read to take note of recurrent themes and convergent ideas. The key issues, concepts, and themes were examined and referenced. Initial codes were generated by writing the applicable codes in the margins of Microsoft word. One of the authors (MAA) cross-validated the lead author's notes made from the transcripts to minimize errors and to ensure the process extracted all significant statements from respondents. This ensured making a better sense of the data and deduce meanings from it. The codes were then searched and grouped into themes and shared with all the authors. All authors discussed the themes and

agreed on those that were recurrent and combined them into broader themes. Those considered important but less recurring were given prominence in the presentation of data. The finally included themes were based on the research objectives as well as those that emerged from the qualitative data itself. Quotes from the respondents or voices of participants have been presented under the various themes anonymously.

## Ethics

The study received ethical approval from the Ghana Health Service Ethics Review Committee with approval number (GHS-ERC 111/05/17). Permission was also sought from the Hohoe Municipal Health Directorate as well as the Hohoe Municipal Hospital Management Team. Furthermore, a written informed consent was sought from each study participant and confidentiality of information and anonymity of subjects were ensured. Respondents had the freedom to participate or quit the study without any harm or discrimination.

## Results

The findings of the study are presented below according to the predisposing, enabling and need factors as conceptualized by Andersen [30]. The predisposing factors consist of the socio-demographic characteristics of respondents, the enabling factors are those motivating access and use of health care services and barriers to health care access and severity. These factors are not mutually exclusive and are interdependent and overlap each other. They are presented under different themes and sub-headings.

### Socio-demographic characteristics of respondents

The socio-demographic characteristics of the study participants in terms of age, marriage status, religion, ethnicity and level of education are presented in Table 1.

### Facilitators and barriers to health care access and utilization among the insured and uninsured

Participants alluded to two major factors as underlining their motivation to seek and utilize health care services. These they referred to as illness severity, symptom persistence and affordability of health care cost.

**Illness severity and symptom persistence.**   Severity of illness and symptom persistence emerged as crucial factors determining the desire to seek and use health care services among both insured and uninsured clients in the Hohoe Municipality. A number of participants expressed their views regarding the two factors as seen below.

*I mostly take drugs from the pharmacy when I'm not well. I only go to the hospital when it becomes serious.*

(35 years old, uninsured female).

*Sometimes I go to the drug store and explain to the pharmacist. When it's worse, I go to the hospital.*

(A 33 year old, insured male).

**Affordability of health care cost.**   Affordability of care was also found as both a major motivational factor in seeking and utilizing health care services among the insured, and a

**Table 1. Socio-demographic characteristics of respondents.**

| Characteristics | Frequency | Percent (%) |
|---|---|---|
| **Age** | | |
| 20–29 | 12 | 23.5 |
| 30–39 | 24 | 39.2 |
| 40–49 | 11 | 21.6 |
| 50–59 | 4 | 7.8 |
| 60+ | 4 | 7.8 |
| **Sex** | | |
| Male | 19 | 37.3 |
| Female | 32 | 62.7 |
| **Marital status** | | |
| Never married | 19 | 29.4 |
| Married | 34 | 66.7 |
| Widowed | 2 | 3.9 |
| **Insurance status** | | |
| Subscribers | 25 | 50.0 |
| Non-subscribers | 25 | 50.0 |
| **Religion** | | |
| Christianity | 49 | 88.2 |
| Islam | 6 | 11.8 |
| **Ethnicity** | | |
| Ewe | 49 | 88.2 |
| Mole-Dagbani | 4 | 7.8 |
| Hausa | 2 | 3.9 |
| **Level of education** | | |
| Primary | 3 | 5.9 |
| JHS/Middle school | 17 | 33.3 |
| SHS | 16 | 23.5 |
| Tertiary | 15 | 29.4 |
| None | 4 | 7.8 |

**Source:** Field data, 2017.

barrier among the uninsured. The issue of affordability of care was noted when study partici-
pants asserted that subscription to the health insurance makes it possible for them to access
healthcare at an affordable cost.

> *When you go to the hospital with health insurance, it reduces your cost. The difference is, at first, even when I'm sick, I don't go at all because of the high amount you have to pay before the doctors attend to you. So, I only go to the drug store (to buy the medicines), but with the insurance, I go to the hospital anytime I feel sick.*

(Insured, male, 22 years).

> *What I think is, as for the drips and the drugs, if I was having the health insurance, the money will reduce. I don't think they'll sell some of the medicines to me. When I was admitted, there was another lady there who had the health insurance so she didn't pay some monies that I paid. So when you're having it, the money you'll pay for the drugs will reduce than if you're not having.*

(Uninsured, female, 23 years).

On the other hand, a major barrier identified among the uninsured regarding health care access and use was lack of health insurance. The uninsured had difficulty in accessing health care because they had no health insurance to cater for the cost of health care services. A 54-year-old uninsured female noted for example; *"I don't go to the hospital because when I reach there and I don't have health insurance and they ask me to buy medicine, it will be very difficult".*

Other respondents noted as follows;

*It was difficult for me because I was admitted for three days and I spent almost 600 Cedis (US $103.46) but those who had it didn't pay as much so if you have the insurance, it's better because though they'll not give you the same things, all that they give is good.*

(Uninsured, male, 28years).

*I sent my brother to the hospital and I spent 70 Cedis (US$12) even with health insurance. The only good thing about the insurance is that you won't pay for folder but if you don't have it, you will pay. In my opinion, it will be difficult for someone who doesn't have health insurance to go to the hospital because of the amount of money they'll pay without it.*

(Insured, Female, 32).

On the barriers to health care access and utilization, the insured identified poor service quality. Shared barriers experienced by both insured and uninsured were: inadequate finances to meet health care needs, poor attitude of service providers, and prolonged waiting time at health facilities. These are presented below.

**Perceived low service quality.** The insured identified perceived low service quality as a major barrier to health care access and use with lamentation about the quality of services they received whenever they visit the health facilities.

*I have been in a position where I didn't have the health insurance. I went to the hospital and I was seen to properly because I could dictate to the people what I wanted them to do and all that. I paid lots of money though. But with the health insurance, I couldn't ask for everything that I wanted.*

(Insured, male, 26 years).

*You'll even go and they have the drugs but they'll tell you they don't have it and they'll still write it for you to go and buy; so why don't you just go and buy your drugs. The only thing they give there is paracetamol and some malaria drugs. They'll sell the other medicines to those who will pay.*

(Insured, female, 32 years).

*At Hohoe (government hospital), about five people will go with different problems but they'll give them the same medicine. It's the same chloroquine, paracetamol and amoxicillin that they give. You'll even worsen your case when you go with it (health insurance). If you want to go with the health insurance, then go around 5:00 am at dawn.*

(Insured, male, 32 years).

Regarding diagnosis and treatment at health facilities, one subscriber remarked:

*And when you go, instead of them to test and examine you, they'll just give you medicines based on what you tell them, meanwhile, in your system you feel that something is wrong but they'll just give you paracetamol. . ..*

(Insured, male, 33 years).

Buttressing the position of the insured, even some of the uninsured alleged the low quality of care, including being prescribed with "low quality drugs", waste time at the hospital, while often times they do not get their drugs at all.

*My hands keep itching and when I went, the medicines they gave me didn't do anything so now when it comes, I just leave it and it goes on its own. So I'm tired of going to the hospital that's why I haven't gone to renew it.*

(Uninsured, female, 34 years).

*What I discovered about the health insurance was that when you're sick and you go and make your complaint, they write the medicines . . .. . .. for you to go and buy. So I see it as a waste of time. You'll go and it will take much time before they'll attend to you and they'll write the medicine for you to go to pharmacy or wherever to buy. So I thought it's wise to just go and make my complaint at the pharmacy so that I won't go and waste my time for inferior services at the hospital. Because of that I didn't renew my insurance when it expired.*

(Non-subscriber, male, 24 years).

Another participant was more critical of the health insurance and quality of care.

*It's not efficient because when you go for treatment and you have cash, they'll treat you better than when you have the insurance. And with cash and carry, sometimes if they want you to do a scan, for instance, they'll ask you to do it but with health insurance, they'll tell you we're out of this, we're out of that. The insurance is almost becoming like a dead trap. You'll go to the facility, and let's say you're diagnosed of typhoid. If you have the insurance, they'll do the first test for you to confirm it. Without the insurance, I'll insist that they do a retest to see the degree of infection which I believe will influence how it should be treated.*

(Uninsured, male, 27years).

**Financial demands for health care.** Financial access to health care was identified as a major barrier among both insured and uninsured. Those who could afford to pay from out-of-pocket were in better position to obtain the care they needed. For the uninsured, utilization of health care services is entirely about money and it is difficult to access health care services without money.

*In my opinion, there is no difference between having and not having the insurance. It's about money. If you don't have money, you cannot go to the hospital even if you have the insurance because when you go, they'll end up writing all the medicines for you to go and buy it. But if you have it, it is good for emergency cases when you don't have money.*

(Uninsured, female, 32 years).

Buttressing the point, a 61 year old insured female had this to say: *"If you don't have money, even with the health insurance, you can't go to the hospital because you'll still have to pay some money when going for your medicines".*

**Attitudes of service providers.** Health service providers' poor attitude towards patients during interpersonal relations and health seeking encounters was also identified as a major barrier to accessing and utilizing services. Participants recounted their negative experiences with health care service providers when seeking health care at the Hohoe Municipal Hospital. As a result, the uninsured and those who are able to afford prefer to visit private health facilities or the pharmacies while some insured prefer to visit other public hospitals outside Hohoe. The following verbatim quotes were largely agreed by majority of uninsured and insured.

*Sometimes, you will go to the hospital and the way the nurses there will be behaving towards you is very bad, especially at the government hospital. They just sit there and chat instead of attending to you. Emergency is emergency oooh!.*

(Uninsured, female, 32 years).

*I go there (Marquart- Margret Marquart Catholic Hospital is located in Kpando in the Volta Region of Ghana) because for there, their patient service is far better than Hohoe hospital. I used to go there (Hohoe hospital) but the treatment has prevented me from going there. Sometimes when you go there, the nurses ignore, as if you're disturbing them. Some even talk to you anyhow. And when you're sick too, after speaking to the doctor, they'll just write the drugs for you without testing you. But at Marquart, they'll attend to you nicely so I developed love for them.*

(Insured, female, 24 years).

The poor attitude, particularly, of the nurses towards clients was further buttressed by a 26-year-old insured female living at close proximity to the Hohoe Municipal Hospital in the following words: "*The hospital is just behind us here but we don't want to go. Some of the nurses are not friendly*".

**Prolonged waiting time.** Added to all the above identified barriers was the issue of prolonged waiting time when accessing and utilizing health care at the Municipal hospital. Both the insured and uninsured expressed frustration and lamentation about the situation, which serves as a demotivating factor in going to the facility for services.

*Sometimes when you go there, you have to wait for a long time in a queue, meanwhile, you have a serious problem.*

(A 26-year-old uninsured male).

*At the hospital, you waste the whole day because clients are many, especially, at the dispensary. And the nurses too, if you're asking them questions at the hospital, they won't mind you. That sometimes discourages me.*

(A 53-year-old insured female).

The prolonged waiting time was, however, reported to be experienced by those who access health care at Government hospitals, and not unique to the Hohoe Municipal Hospital. The following quotes summarize this claim.

*The person without it (health insurance) will have advantage more than the one who is having because if you have money and you go to a private hospital, they'll treat you well and faster. People with it normally go to the government hospital where there is so much delay.*

(Insured, female, 24).

Corroborating this position, an interviewee recounted her experience during her last visit to the hospital:

*I used to go to the government hospital. I wasted my whole day there but I didn't get any better medicine and I didn't feel better so I started going to one pharmacy.*

(48-year-old uninsured female).

The experience and feelings were not particularly exclusive to either the insured or uninsured as the former also indicated prolonged waiting. In their opinion, however, they attributed it to time used in processing their insurance.

*You'll take your card to the OPD, after which you'll be asked to wait till they process the health insurance forms for you. Someone without the health insurance will not wait for this but they will have to pay more because they are now going to pay for a folder and other things.*

(Insured, male, 39 years).

However, healthcare providers had a word of advice to avoid waiting for long hours.

*When you come early, you will leave early because health care provision is based on first come first serve. Now some prescribers report by 5:30 or 6:00 am. Those who come early from 5:00 am to 7:00 am, they'll leave early but if you come late and you expect a nurse to let you cross (jump the queue) the one who came earlier than you, then you lie bad, unless your condition is critical before we cross you or we send you to emergency. So normally those who say that came late and they expected us to cross them.*

(OPD nurse, Female, 32 years).

This situation the providers also explained often lead to some clients sometimes being impatient and disrespectful simply because of the feeling of entitlement due to health insurance card ownership which makes their work very difficult when providing care.

*We have challenges, especially, the lack of understanding of the patient's impatience and one thing they do not understand is that everything is a process. So when they come to our side, they feel we have not attended to them and they get angry, send our names to the radio station saying we are not active because nowadays you have to enter everything into a database.*

(Dispenser, Female, and 26 years).

### Service providers views and experiences of providing care to NHIS subscribers and non-subscribers

Health care providers' views about the insured and uninsured were sought in two broad areas. Firstly, their views on the experiences of subscribers and non-subscribers in accessing and utilizing health care services. Secondly, their experiences in providing health care services under the NHIS and out-of-pocket payments made by uninsured clients for accessing health care. Beyond the two broad areas, providers were also asked of their perception as to the extent the NHIS has made a difference in equitable access to health care in the municipality.

Analysis of the in-depth interviews of health care service providers revealed how it has made service provision easier for the insured compared to the uninsured. In their opinion, there is

easy access to formal health care services for the insured while the uninsured find it difficult to utilize health care and in most cases report only with emergencies due to financial constraints.

With regards to their experiences of providing health care services under the NHIS and out-of-pocket payments, they indicated that comparatively, it is easier with the insured, for payment is already taken care of. Thus, they are able to make sound decisions and provide the needed care for subscribers without worrying about cost.

> *Oh it's easy with the health insurance. Some of them when they come without health insurance, their condition deserves admission but because they don't have health insurance and they don't have money too, it's very difficult to provide service to them and you can't also ask them to leave with that condition so it's easy if they have the health insurance.*

(OPD nurse, Female, 32 years).

On issues of equity and allegations of preferential treatment for cash paying, out-of-pocket paying or the uninsured, service providers opined that they treat their clients equally without discrimination. The only exception being that the uninsured have to pay out-of-pocket for health care services as expressed in the following quotes.

> *When you come to the hospital or health facilities, we don't know whether you have health insurance or not unless you get here before we will know that you don't have health insurance. So I can't tell that you are not having health insurance so this is your queue. There is no announcement that will be made that this is the queue for those who are insured and this is the queue for those who are not insured.*

(OPD nurse, Male, 30 years).

> *Some of them when they come to hospital, especially, those who are not insured and they experience the cost that they bear, they will know that the health insurance is important because . . . . . . . . . . . . . . . .. whatever that we do for you, checking your temperature, even if it is injection, . . . . . . . . .., you will pay everything. The bed that you sleep on too, you'll pay everything. Those with insurance are not paying but if you don't have insurance, everything that they do for you, you'll pay. . ..*

(OPD nurse, female, 32 years).

One concern was about inequitable access to free basic health care for all regardless of insurance status.

> *Yh it (health insurance) has helped but me what I normally say is that in the Constitution they wrote there boldly, I have forgotten the clause but it says that free basic health care for everyone and they didn't say that to get the free basic health care, you have to get insurance. So every Ghanaian regardless of whether you have insurance or not, you have to get the free basic health care. In this case, you may only have to come here (municipal hospital) as referral.*

(OPD nurse, Male, 30 years).

### Reasons for non-subscription and recommendations to make the NHIS better

Several reasons or explanations were advanced by participants as to why some people are not subscribed to the NHIS and why others have dropped out of the scheme. Reasons for non-

subscription and drop-outs by the uninsured included lack of resources for subscription, long waiting hours and perceived substandard care associated with NHIS already discussed above. Other reasons included low NHIS coverage, infrequent episodes of illness, and NHIA administrative issues.

Regarding lack of resources as a reason for non-subscription, a 54-year old male uninsured for instance, noted: *"I spent a lot of money because I didn't have health insurance so after that encounter (operation), I have decided to have it but I am looking for money now"*.

Participants also expressed concerns about not falling sick regularly and so their subscription to the scheme will be a waste of resources as they asserted:

*The last time I went to the hospital, I had an accident and I paid for everything. Before then, I haven't been to the hospital for about five years so even if I want to add up the NHIS premiums I would have paid for those five years, it's a lot of money.*

(Uninsured, male, 27 years).

*I don't fall sick so I'll just be wasting money. This house, for instance, the children are many, if their father doesn't get money for renewal, it is very difficult to renew. They all have it but haven't renewed because they don't fall sick.*

(Uninsured, female, 32 years).

Although the NHIS covers about 95% of the common illnesses affecting Ghanaian residents, participants expressed their disappointments in having to pay out-of-pocket most of the time despite having health insurance.

*For me, I think the health insurance has come to help but it is not properly designed because for now, excuse me to say, there are some drugs that I'm a business man so I can buy but it's covering what I can afford and not what I cannot afford. I've not really benefited as such but in one way or another, it helps.*

(Insured, male, 39 years).

*The first one (initial implementation) was better but this time, the medicines you'll buy are too much. You even buy medicines that are covered by the insurance because they'll tell you it's finished.*

(A 22-year old insured female).

Some of the participants also attributed their uninsured status to administrative issues of the NHIA. A 35-year old uninsured female remarked:*"I've been there (registration office) a couple of times and they always tell me the network is bad, so going back there again is difficult but I still want to have it"*.

## Recommendations

Participants also made a number of recommendations to make the NHIS better including a reduction in NHIS premiums, expansion of services covered under the scheme, and motivation of health care providers to develop positive attitudes towards clients, especially, the insured. A number of interviewees provided suggestions which are stated below verbatim.

*The renewal fee too is expensive for me, so if they can reduce it or make it free, maybe I'll go and register again.*

(A 34-year old uninsured female).

*They should make the health insurance cover all the medicines so that even if we spend the whole day at the hospital, we know we can get all our medicines.*

(A 24 year old uninsured male).

Participants appealed to the NHIA to speak to the health care providers to behave nicely towards the clients, especially the insured. A 38 years old insured male noted: *"they should speak to the nurses on their conduct towards those who use the health insurance to motivate more people to use it"*. For their part, health care providers also suggested the need for the NHIA to educate clients to be patient with care providers while they try to provide focused care which often takes time, yet meet the needs and expectations of clients.

*Early morning, come and educate them that when they come they should have patience for us because we are also human beings. Most of them think that it is because of them we are here so they can talk to us anyhow. When I insult or quarrel with you, will you have the free heart to do something for me? Both of us need patience and self-control but they think it's only us who need that self-control and patience but we are all humans.*

(OPD nurse, Female, 32 years).

## Discussion

The paper provides empirical evidence of the experiences of the insured members of the NHIS and the uninsured in accessing and utilizing health care services in the Hohoe Municipality. These experiences are complemented by the views of health professionals. Overall, the three category of respondents identified factors motivating health care access and use, barriers to accessing health care, inequity in health care access and utilization between the rich and the poor and the reasons for non-subscription to the NHIS.

### Factors motivating health care access and use among NHIS insured and uninsured

It was found that factors motivating access to and use of health care services among both NHIS insured and uninsured were illness severity and symptom persistence. This finding is consistent with findings of Krumkamp et al., who identified factors that influence health care utilization behavior for children with mild or severe symptoms and found that the effects of other factors such as distance became less important with increasing severity of symptoms, indicating that barriers to utilizing health care facilities are less important if a person presents with a more severe illness [39]. The health insurance also enhanced health care access by reducing cost of health care for the insured which is consistent with arguments by Akum who claimed that the NHIS is beneficial as it serves as a means of using health care services for the insured [18]. Again, responses given by participants concerning utilization of health services only in times of symptom persistence and severity is consistent with the Health Care Service Utilization Model by Andersen, who pointed out that perceived need of the individual, considering their health needs or illness condition, will generate the need for utilizing health services [30].

## Barriers to health care access and utilization by NHIS insured and uninsured

Non-ownership of health insurance was found as a major barrier to accessing healthcare among the uninsured which confirms findings of Gobah and Zhang that lack of insurance is the single most important reason for not seeking formal care among the non-insured, with a higher proportion of the uninsured seeking late or delayed care due to their non-insurance status [40]. This is also consistent with Andersen argument that having health insurance or health insurance status is not only an important determinant but also influences utilization of health services [30]. This study finding also corroborates findings from Fenny et al which showed that health insurance status, education, and gender, were the three main determinants of health care utilization [41].

Perceived low service quality was also identified as a major barrier for the insured. This was reported to come in the form of unfriendly attitude from health service providers, writing prescription without carrying out laboratory investigations, nurses ignoring them as if they were disturbing them and long waiting time at the OPD and dispensary or pharmacy among others. This is consistent with an earlier study by Akum who found that subscribers perceived verbal abuses, long waiting times, not being physically assessed and being discriminated against in favor of the uninsured and the rich because providers think that the insured were abusing their services by frequenting the facilities, and sometimes pretending to be sick to collect drugs for their uninsured relatives [18]. Other perceived low service quality in the study included subscribers also having to spend more waiting time, suffer verbal abuses for using NHIS card rather than cash, and paying extra money to cover for extra drugs and services that are not covered under the scheme [18].

The study also found that meeting financial demands to accessing health care was a challenge for both NHIS insured and uninsured. This finding is consistent with findings of Sulemana et al., who found that access to health services in deprived communities is influenced by inadequate health facilities, long distance to health facilities, and inability to afford the cost of health services [42]. The insured are also challenged with extra charges when seeking health care. This finding is also consistent with findings of Acheampong who found that healthcare expenditure not being covered by the NHIS, additional payments when holding NHIS card, and inaccessibility of needed care also affected the utilization of health care services with health insurance [43].

Attitude of service providers at government healthcare facilities, especially, was also found to be poor. Again, this manifested in the form of complaints about what respondents described as rude and impolite attitude of nurses and at public health facilities compared to private ones. This finding is in agreement with the findings of Ganle et al., who evidenced that inequitable distribution of maternal health care services, intimidation experienced in healthcare facilities, unfriendly health care providers, cultural insensitivity, long waiting hours, lack of privacy, and poor care quality were found to be important barriers to access and use of health services in Ghana [44]. This finding is critical as Andersen suggested in the Health Care Service Utilization Model that people will utilize health care facilities if they believe that the facility is effective in addressing their health needs [30]. Patients who feel intimidated will most likely lose the confidence to be open with their disease condition, resulting in health care needs not effectively addressed. When this happens, access and utilization of health services will remain poor thereby affecting Ghana's efforts at attaining UHC.

This, notwithstanding, views of service providers regarding their experiences in providing care to both insured and uninsured were more positive with the former compared with the latter. This may be the reason why non-subscribers are being asked why they do not have the

health insurance when they report to the health facility, which to them is embarrassing. This finding also shows that the health insurance provides financial protection to subscribers, enhancing their access to formal health care when they need it which Andersen describes as an enabling factor [30].

Prolonged waiting time was also found as a barrier to health care access by both NHIS insured and uninsured. This is consistent with findings of Alhassan et al who evidenced that increased pressure from high demand of health care services at health facilities for insured clients resulted in longer waiting times, illegal charges, and violation of professional conducts by service providers [24].

## Inequitable access to health care between the rich and the poor

In comparing views and expressions among the insured and uninsured in accessing and utilizing health care services, the study found an inequity in the access and use of health care services on the basis of financial status. For instance, prolonged waiting time when accessing health care and poor attitude of health service providers were barriers experienced by both NHIS insured and uninsured who utilized government hospitals. Those who had the financial means to access healthcare from private facilities were free from waiting and poor attitude of service providers.

Again, with regards to quality of services and access to drugs as expressed by study participants, the study found that the rich were advantaged. The rich are able to pay for extra charges, afford private care services, and pay for prescribed drugs regardless of their insurance status.

The insured reported that the health insurance helps to reduce the cost of health care services, making accessibility relatively easier for the insured but due to extra charges, those who cannot afford prescribed medicines which are not covered by the NHIS are thus disadvantaged.

This finding confirms findings of Noi who evidenced that insured clients are unwilling to access health care with their NHIS cards due to delays associated with health care utilization with the cards [20]. Other insured would also prefer to attend private health facilities where they even pay higher fees out-of-pocket to attain quality health care services [20]. This finding also confirms findings of Abuosi et al., and Alhassan et al., who found that despite the NHIS causing an increase in health service utilization in Ghana, there had been concerns about the quality of care under the scheme [24,45]. These concerns range from long queues, waiting times, verbal abuse of patients by service providers, quality of drugs, and inadequate physical examination by doctors [24,45]. The level of inequity associated with access to health care services between the rich and the poor also buttresses findings of Ganle et al., who found a marginal increase in access to and utilization of maternal health services following a user fee exemption policy [44]. However, large gradients of inequities exist because more women in the highest wealth quintile than women in the lowest accessed and used all components of skilled maternal health services. This raises questions about the potential equity and distributional benefits of Ghana's user-fee exemption policy, and the role of non-financial barriers [44].

Relating this study finding to the conceptual framework by Andersen [30], it can be emphasized that the presence of a health care facility, travel time to the facility, cost involved in accessing care, health insurance status, and how effective this facility is in addressing their health needs, greatly influences general population use of health facilities. It can be asserted from this study finding that even with the health insurance, some of the subscribers who could not afford extra charges are not able to access health care. Furthermore, those who could afford to pay at private facilities sought health care from there because of their perceptions that their health needs will be effectively addressed over there.

### Reasons for non-subscription to the national health insurance scheme

Reasons for non-subscription as noted by study participants included lack of resources for subscription, long waiting hours and perceived substandard care associated with NHIS, low NHIS coverage, infrequent illness episodes, and NHIA administrative issues. These reasons were said to potentially discourage people from subscribing to the scheme. Other findings have also reported similar barriers to NHIS subscription. Boateng et al, for instance, noted that poor service quality, lack of money, and taste of other sources of care, as well as self-rated health influenced non-renewal of NHIS subscription [46]. Regarding low NHIS coverage of some specific conditions which participants alluded to in this study, Alhassan and colleagues also noted in their study that while the NHIS covers common diseases such as malaria, upper respiratory tract infections and diarrheal diseases, relatively uncommon diseases such as cancers, are not covered by the NHIS [24]. Kumi-kyereme et al also found the major barriers to NHIS subscription to be long queues and waiting time, perceived poor quality of drugs, and negative attitude of service providers both at the healthcare facilities and the health insurance office [17].

In relation to the conceptual framework, Andersen mentioned that the external factors that are important in influencing an individual's decision to use health care services, in this case subscription to the NHIS, includes the presence of a health care facility, travel distance to health facility, cost involved in accessing care, health insurance status, and how effective this health facility addresses individual health care needs [30]. The cost involved in subscribing to the scheme for non-subscribers and the perception that the NHIS is not effective in addressing their health needs due to the challenges associated with it had influenced their decision to remain uninsured. Also perceptions of people concerning health care services quality, influences health service utilization as well [29].

## Study limitation

The limitation of this study is that out of the seven sub-municipalities in the Hohoe Municipality, this study was conducted in only one sub-municipality using a relatively small sample size due to its qualitative nature and thereby making it impossible to generalize the findings. The reliability and validity of the study is, however, not limited in anyway.

## Conclusions

Wealth status still remains a strong determinant to health care access and use even with the NHIS. The uninsured face a major challenge of financing their health care cost while the insured do not have all their healthcare expenditures covered despite having health insurance. The implication of financial barriers to health care access is that the poor will not be able to access health care which questions the efficiency of the NHIS in ensuring equity and removing financial barrier to health care access. If this persists, Ghana may not be able to meet the Sustainable Development Goals of ensuring healthy lives and promoting well-being for all at all ages and UHC by the year 2030. It is recommended that the Hohoe Municipal Assembly appeal for funds from corporate organisations, religious institutions, opinion leaders, and traditional rulers in the Municipality to raise enough funds to support subsidized premiums and registration fees for non-working population in the municipality to curb the barrier to NHIS subscription. The Government should consider expanding coverage of the health insurance to cover some of the major health problems so as to bridge the financial inequity with health care access between the rich and the poor. This could be achieved if the government considers UK health service approach where services are hundred percent financed from central taxation. But this also presents a challenge with the narrow tax base but creating a reliable and up-to-

date database of all businesses in the formal and informal sectors for tax purposes will help. Implementing these recommendations may improve access to and use of health care services, as well as subscription to the NHIS, thereby improving equitable access to and utilization of health care services in the Hohoe Municipality.

## Supporting information

**S1 File.**
(DOCX)

## Acknowledgments

We thank all the people and hospital staff who participated in the in-depth interviews.

## Author Contributions

**Conceptualization:** Suraiya Umar.

**Data curation:** Suraiya Umar.

**Formal analysis:** Suraiya Umar, Adam Fusheini.

**Investigation:** Suraiya Umar, Martin Amogre Ayanore.

**Methodology:** Suraiya Umar, Adam Fusheini, Martin Amogre Ayanore.

**Resources:** Suraiya Umar, Adam Fusheini, Martin Amogre Ayanore.

**Supervision:** Adam Fusheini, Martin Amogre Ayanore.

**Validation:** Adam Fusheini, Martin Amogre Ayanore.

**Writing – original draft:** Suraiya Umar.

**Writing – review & editing:** Adam Fusheini, Martin Amogre Ayanore.

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
