## [Decision Letter · Decision Letter 0]

24 Sep 2020

PONE-D-20-18400

The shared experiences of insured Members and the uninsured under Ghana's National Health Insurance Scheme:  Evidence from the Hohoe Municipality

PLOS ONE

Dear Dr. Fusheini,

Thank you for submitting your manuscript to PLOS ONE. After careful consideration, we feel that it has merit but does not fully meet PLOS ONE’s publication criteria as it currently stands. Therefore, we invite you to submit a revised version of the manuscript that addresses the points raised during the review process.

The manuscript has been evaluated by four reviewers, and their comments are available below. Overall, the reviewers provided positive comments about your study, but they have also raised a number of concerns (especially regarding the methodology reporting) that need attention. Could you please revise the manuscript to carefully address all the concerns raised?

We look forward to receiving your revised manuscript.

Kind regards,

Dario Ummarino, Ph.D.

Associate Editor

PLOS ONE

Journal Requirements:

2. Please include additional information regarding the survey or interview guide used in the study and ensure that you have provided sufficient details that others could replicate the analyses. For instance, if you developed a questionnaire as part of this study and it is not under a copyright more restrictive than CC-BY, please include a copy, in both the original language and English, as Supporting Information.

3.We note that "Map of Hohoe Municipal" in your submission contain [map/satellite] images which may be copyrighted. All PLOS content is published under the Creative Commons Attribution License (CC BY 4.0), which means that the manuscript, images, and Supporting Information files will be freely available online, and any third party is permitted to access, download, copy, distribute, and use these materials in any way, even commercially, with proper attribution. For these reasons, we cannot publish previously copyrighted maps or satellite images created using proprietary data, such as Google software (Google Maps, Street View, and Earth). For more information, see our copyright guidelines: http://journals.plos.org/plosone/s/licenses-and-copyright.

1.    You may seek permission from the original copyright holder of "Map of Hohoe Municipal" to publish the content specifically under the CC BY 4.0 license. 

Reviewers' comments:

Reviewer's Responses to Questions

**Comments to the Author**

1. Is the manuscript technically sound, and do the data support the conclusions?

Reviewer #1: Yes

Reviewer #2: Yes

Reviewer #3: Yes

Reviewer #4: Partly

2. Has the statistical analysis been performed appropriately and rigorously? 

Reviewer #1: N/A

Reviewer #2: Yes

Reviewer #3: N/A

Reviewer #4: N/A

3. Have the authors made all data underlying the findings in their manuscript fully available?

Reviewer #1: Yes

Reviewer #2: Yes

Reviewer #3: Yes

Reviewer #4: No

4. Is the manuscript presented in an intelligible fashion and written in standard English?

Reviewer #1: Yes

Reviewer #2: Yes

Reviewer #3: Yes

Reviewer #4: Yes

5. Review Comments to the Author

Reviewer #1: Very important study of the lived experiences of the insured and the uninsured with access to and use of healthcare services in Ghana; also interesting and very well-written.

Here are few recommendations to address:

1) Under the introduction, please note that not all developed countries "have successfully ensured optimum levels of healthcare for their entire populace," as has been stated, an example of this is the United States where a large proportion of the population is still uninsured and so have no usual access to care.

2) "bedevil" not an appropriate term to use in a technical writing as in this manuscript.

3) page 8: Why are the % in parenthesis? eg. about (55%)??

4) (25%) ---Since this figure appears at the beginning of a sentence, the accepted norm is to write it out in words.

Reviewer #2: The study is relevant most especially at this moment of economic crisis when out-of-pocket payment for health care services might be a burden for the uninsured. It relevant to the SDG G-3 which proposed universal health coverage for all by the year 2030. I think the manuscript will contribute positively to the rating of this journal.

Reviewer #3: Thank you for your submission. I find the topic of interest and importance. Additionally, it is overall quite well written with beautiful vocabulary.

I have several minor suggestions which I have detailed below:

- On page 1, I would recommend including the word 'some' in the second paragraph to state, "While some developed countries have successfully ensured optimum levels of healthcare for their entire populace,..."

-On page 3, you use the abbreviation GHS but I do not see this abbreviation defined previously in the text. Though this may have been an oversight on my part, I could not find it. You also translate this currency into $ but not define which dollar you are referring to... United States of America (US dollars) or another dollar?

- On this same page you refer to the indigent or core poor. It may be helpful to specifically define these terms. Furthermore, it could be of interest to touch about what percentage of the Ghanian population is considered indigent or core poor.

-On page 4, you use the term 'rare illness episodes'. I assume here you are referring to rare episodes of illness, rephrasing this may help avoid any confusion, as this could upon first glance be seen as referring to illnesses that are rare or uncommon, such as cancers. Though please note this is more of a style suggestion and this change is not essential as I think the point is well understand via the context.

- One page 12, I would use remain consistent and use the term "symptom persistence" both in the body of hte first paragraph and the title of the next. I also feel "symptom persistence" sounds more natural in English.

- On page 24 under conclusions, the following sentence needs a few corrections: "The uninsured face a major challenge of financing their health care cost, while (not whiles) the insured do not...". Moreover I would not use the pronoun "her" to refer to Ghana. "If this persists, Ghana may not be able to meet its..."

-Under acknowledgements I would more directly state, "We thank all the people and hospital staff..."

-I feel some parts of the text were fairly repetitive. I might review to see where I could cut out repeated comments and replace the repeated points with sections where you could suggest or explore solutions to all the problems well-detailed in the article regarding the failure of the NHIS system to serve its intended purpose.

- I would be sure to mention setbacks as well. Discuss briefly sample size, biases, etc.

Overall, I felt the work was quite well done and is of value to the research world. I would simply recommend some minor grammatical changes as well as adding some elements regarding setbacks and recommendations or suggestions for potential solutions and/or further research.

Reviewer #4: The findings from this study are interesting and relevant to anyone considering setting up a healthcare insurance programme. Whilst on the surface it would appear that a health insurance programme would create more equitable access to healthcare, the authors of this paper have actually identified several barriers to this goal (e.g. additional costs of drugs over and above the cost of the health insurance, sub-standard care for those with health insurance) which require further consideration. Whilst the methods used in this study are appropriate, more detail needs to be included about how the analysis was undertaken and the qualitative themes presented in the results require further work to ensure the main findings are easily identifiable. Below I have outlined some specific examples about how the write-up of this paper could be improved:

Introduction

The introduction is quite lengthy and could be more focused.

In the 4th paragraph of the introduction the authors discuss the NHIS and the potential problems with this. In the subsequent paragraphs they explain what the NHIS is. The introduction would be greatly improved if they explained what the NHIS is before discussing the problems.

Anderson healthcare utilisation model: This was discussed during the introduction but was not mentioned anywhere else in the paper and so it is unclear what relevance this has to this particular study? If the authors are keen to use this model in their study they need to provide more information about how their data collection was developed using this model and whether or not the results align to this model.

Aims and objectives: It would be useful if the authors could include a clear statement about the aims and objectives at the end of the introduction. E.g. “the purpose of this study was to compare the experiences of NHIS insured and uninsured in accessing and utilizing healthcare services. Specifically we aimed to….”

Methods

The authors state that they chose five communities from within the Hohoe-sub municipality. Why were these five communities and are they representative (e.g. different socioeconomic communities) of the wider municipality?

How were participants recruited? Gatekeeper? Posters? Online advertising?

The authors should provide specific information about what topics were covered in the interview schedule.

More detail is required about how the analysis was conducted / how the themes were decided upon. E.g. How many of the authors conducted the analysis? Were the themes checked and agreed on by multiple authors?

Results

The themes aren’t clearly defined, with a lot of overlap between them.

The authors split the themes into factors motivating health care access and barriers to healthcare access. However, affordability of health care cost appears to be both a barrier and a facilitator and by splitting the discussion between the two themes the importance of this finding seems to get lost. Perhaps the authors may consider discussing barriers and facilitators within themes rather than having them as the main themes. E.g. create a theme “affordability of health care cost” and then discuss the barriers and facilitators of this as well as the differences / similarities between the insured and uninsured.

Many of the issues discussed under “reasons for non-subscription and recommendations…” were discussed under other headings so this doesn’t seem like a separate theme. As mentioned above, these issues can be discussed under more specific themes.

In the methods the authors mention that health workers were interviewed but there is no further information about what they were interviewed about and there is no mention of this in the results?

Discussion

At the start of the discussion the authors mention views of health professionals but these weren’t actually discussed in the results.

6. PLOS authors have the option to publish the peer review history of their article (what does this mean?). If published, this will include your full peer review and any attached files.

Reviewer #1: **Yes: **Dr. Joyce Addo-Atuah

Reviewer #2: No

Reviewer #3: **Yes: **Caroline Barnes

Reviewer #4: No

---

## [Author Response · Author response to Decision Letter 0]

19 Oct 2020

Dear Editor-In-Chief,

Subject: Revised submission of manuscript for consideration 

We found the reviewers comments to be very helpful in guiding us through what became an extensive revision process of a ‘minor revision’. While the main objectives of writing the article remain intact, there has been some few movements of paragraphs due to the reviewers’ comments. We also acknowledge that much of the text on health providers’ views is new in order to meet the suggestions of one of the reviewers. We are grateful for reviewing our paper and providing us with the opportunity to resubmit the revised manuscript. Enclosed to this letter is our revised manuscript titled: “The shared experiences of insured members and the uninsured in health care access and utilization under Ghana's National Health Insurance Scheme: Evidence from the Hohoe Municipality”.

In revising the manuscript, we took into account all reviewers’ suggestions and constructive criticisms of our initial submission. A point-by-point explanation of how we addressed the comments raised by each reviewer is detailed below. We believe that the reviewers’ comments helped us to make a critical revision of the manuscript, which will be interesting for readers in the wider scientific community. 

All authors have read and approved this revised manuscript, and agreed with the publication of their names. There are no conflicts of interest. 

We look forward to hearing from you on this matter.

Sincerely,

Adam Fusheini (PhD)

(Corresponding Author)

Detail Revision notes on revised Manuscript

Manuscript ID: PONE-D-20-18400

Title: The shared experiences of insured members and the uninsured in health care access and utilization under Ghana's National Health Insurance Scheme: Evidence from the Hohoe Municipality

Below, we explain how we have addressed each of the comments of the reviewers.

Journal Requirements

Response: We have reformatted the manuscript in line with PLOS ONE’s style

2. Please include additional information regarding the survey or interview guide used in the study and ensure that you have provided sufficient details that others could replicate the analyses. For instance, if you developed a questionnaire as part of this study and it is not under a copyright more restrictive than CC-BY, please include a copy, in both the original language and English, as Supporting Information

Response: Additional information is provided on the interview guide used on pages 10-11, lines 241-249 of the revised/tracked changes version. Also, a copy of the interview guide has been included as supporting information.

3. We note that "Map of Hohoe Municipal" in your submission contain [map/satellite] images which may be copyrighted.

Response: The “map of Hohoe municipal” has been removed from the revised manuscript.

REVIEWER #1:

Comment 1: Under the introduction, please note that not all developed countries "have successfully ensured optimum levels of healthcare for their entire populace," as has been stated, an example of this is the United States where a large proportion of the population is still uninsured and so have no usual access to care.

Authors’ response: The sentence has been corrected and “some” has been inserted to indicate that not all developed countries have successfully ensured optimum healthcare for their entire population. The sentence now reads as:

“While some developed countries have successfully ensured…” in line 17 of page 1 of the tracked changes version of the manuscript.

Comment 2: "bedevil" not an appropriate term to use in a technical writing as in this manuscript.

Authors’ response: “bedevil” has been replaced with “face”.

“Health care accessibility and utilization remains limited as a result of financial and socio-cultural challenges that face such economies” in line 20 on page 1 of the tracked changes version of the manuscript.

Comment 3: Why are the % in parenthesis? E.g. about (55%)?

Authors’ response: The wrongly placed parentheses have been removed. The text now reads as: 

“About 55% of the population are engaged in farming. Twenty-five percent are also involved in trading, 15% in livestock rearing and 5% engage in other…” in lines 203-205 on page 9 of the tracked changes version of the manuscript.

Comment 4: (25%) ---Since this figure appears at the beginning of a sentence, the accepted norm is to write it out in words.

Authors’ response: The figure has been written in words as follows:

“Twenty-five percent are also involved in trading, 15% in livestock rearing and 5% engage in other” in lines 204-205 of the tracked changes version of the manuscript.

REVIEWER #3

Comment 1: On page 1, I would recommend including the word 'some' in the second paragraph to state, "While some developed countries have successfully ensured optimum levels of healthcare for their entire populace ..."

Authors’ response: The sentence has already been corrected in line 17 of page 1 of the tracked changes version of the manuscript

Comment 2: On page 3, you use the abbreviation GHS but I do not see this abbreviation defined previously in the text. Though this may have been an oversight on my part, I could not find it. You also translate this currency into $ but not define which dollar you are referring to... United States of America (US dollars) or another dollar?

Authors’ response: The abbreviation GHS has been defined and the currency conversion to dollar has been specified as United States of America dollar (US dollar) as follows:

“The annual premium varies between twenty-five Ghana Cedis (GHS25) (approximately US$4.30) and GHS46 (US$7.91) between the low- and upper- income groups. Those who contribute to SSNIT only need to pay a registration fee of about GHS5 (US$0.86).” in lines 42-44 on page 2 of the tracked changes version of the manuscript

Comment 3: - On this same page you refer to the indigent or core poor. It may be helpful to specifically define these terms. Furthermore, it could be of interest to touch about what percentage of the Ghanaian population is considered indigent or core poor.

Authors’ response: The term “indigent” has been defined in-text and a statistic for the proportion of Ghanaian populace considered as indigent/core poor has been given. The new text reads as:

“…the indigent or extreme poor, estimated to be about 8.2 percent of Ghana’s population in 2016/17 [11] and those identified by the minster for employment and social welfare as needing support [12]. The indigent as defined by Act 650 (2003), which has been repealed by Act 852 of 2012 refers to “a person who does not have a visible or adequate means of income or who does not have a person to support him or her”. See lines 47-51 on pages 2-3 of the tracked changes version of the manuscript

Comment 4: -On page 4, you use the term 'rare illness episodes'. I assume here you are referring to rare episodes of illness, rephrasing this may help avoid any confusion, as this could upon first glance be seen as referring to illnesses that are rare or uncommon, such as cancers. Though please note this is more of a style suggestion and this change is not essential as I think the point is well understand via the context.

Authors’ response: We have replaced “rare illness episodes” with “rare episodes of illness” as suggested. The sentence is now:

“Other reasons included rare episodes of illness, limited benefits” in line 75, page 4 of the tracked changes version of the manuscript

Comment 5: - One page 12, I would use remain consistent and use the term "symptom persistence" both in the body of the first paragraph and the title of the next. I also feel "symptom persistence" sounds more natural in English.

Authors’ response: “Symptom persistence” has been substituted into the text as it reads and sounds better. The texts now read as:

“These they referred to as illness severity, symptom persistence and affordability of health care cost.”

“Severity of illness and symptom persistence emerged as crucial” in lines 320 and 322 on page 14 of the tracked changes version of the manuscript.

Comment 6: - On page 24 under conclusions, the following sentence needs a few corrections: "The uninsured face a major challenge of financing their health care cost, while (not whiles) the insured do not...". Moreover, I would not use the pronoun "her" to refer to Ghana. "If this persists, Ghana may not be able to meet its..."

Authors’ response: “Whiles” has been replaced by “while” and the text now reads as: 

“The uninsured face a major challenge of financing their health care cost while the insured do not have all their healthcare expenditures covered”

Additionally, in the subsequent sentence, we have replaced “her” with “the”:

“If this persists, Ghana may not be able to meet the sustainable development goals of ensuring”. See lines 845 and 849 on page 31 of the tracked changes version of the manuscript.

Comment 7: -Under acknowledgements I would more directly state, "We thank all the people and hospital staff..."

Authors’ response: The statement of acknowledgments has been rewritten as “We thank all the people and hospital staff who participated in the study.” See line 870 on page 32 of the tracked changes version of the manuscript.

Comment 8: -I feel some parts of the text were fairly repetitive. I might review to see where I could cut out repeated comments and replace the repeated points with sections where you could suggest or explore solutions to all the problems well-detailed in the article regarding the failure of the NHIS system to serve its intended purpose.

Authors’ response: We have reviewed and cut down on repeated text.

Comment 9: - I would be sure to mention setbacks as well. Discuss briefly sample size, biases, etc.

Authors’ response: We have stated study limitations on page 31 of the revised manuscript. We have briefly discussed sample size on pages 9 and 10 of the tracked changes version of the manuscript.

REVIEWER # 4:

Comment 1: 

The introduction is quite lengthy and could be more focused.

Authors’ response: Superfluous information have been removed to focus the scope of the introduction.

Comment 2: In the 4th paragraph of the introduction the authors discuss the NHIS and the potential problems with this. In the subsequent paragraphs they explain what the NHIS is. The introduction would be greatly improved if they explained what the NHIS is before discussing the problems.

Authors’ response: Thank you very much for this useful and critical observation. We have now reorganize and restructure the introduction to explain what the NHIS is before discussing the problems as seen on pages 1-6 of the tracked changes version of the manuscript.

Comment 3: Anderson healthcare utilisation model: This was discussed during the introduction but was not mentioned anywhere else in the paper and so it is unclear what relevance this has to this particular study? If the authors are keen to use this model in their study, they need to provide more information about how their data collection was developed using this model and whether or not the results align to this model.

Authors’ response: The results and discussion sections has been modified to reflect the Andersen model of health care utilization. In introducing the results section, we have reworded it as:

“The findings of the study are presented below according to the predisposing, enabling and need factors as conceptualized by Andersen [30]. The predisposing factors consists of…”. See lines 298-301 on page 13 of the tracked changes version of the manuscript. Also, see lines 707-710, 718-719 on page 26; 751-762; 802-809, and 827-835 on pages 27-31 of the tracked changes version of the manuscript.

Comment 4: Aims and objectives: It would be useful if the authors could include a clear statement about the aims and objectives at the end of the introduction. E.g. “the purpose of this study was to compare the experiences of NHIS insured and uninsured in accessing and utilizing healthcare services. Specifically, we aimed to….”

Authors’ response: The aims and objectives have been incorporated into the text as follows:

The purpose of this study was to compare the experiences of NHIS insured and uninsured in accessing and utilizing healthcare services. Specifically, we aimed to explore factors motivating accessibility and utilization of health care services among NHIS subscribers and non-subscribers in the Hohoe Municipality; to assess barriers to accessibility and utilization of health care services by NHIS subscribers and non-subscribers. The study also sought to ascertain possible disparities in health care services provided to NHIS subscribers and non- subscribers; and to assess the views of service providers on the experiences of NHIS subscribers and non-subscribers in accessing and utilizing health care services. See lines 126-133 on page 6 of the tracked changes version of the manuscript.

Comment 5: The authors state that they chose five communities from within the Hohoe-sub municipality. Why were these five communities and are they representative (e.g. different socioeconomic communities) of the wider municipality? 

Authors’ response: the reason for selecting those five communities has been given as follows:

The selection of the five communities was also informed by the cosmopolitan nature of the municipality with different ethnic groups, socio-economic activities and demographic characteristics. This was, therefore, to give a cross-sectional representation of the wider Hohoe communities. See lines 217-221 on page 9 of the tracked changes version of the manuscript.

Comment 6: The authors should provide specific information about what topics were covered in the interview schedule.

Authors’ response: Specific information regarding the main topics of the interview has been given. The text reads as:

The interview guide captured the socio-demographic characteristics of study participants, the motivating factors and or the extent to which the health insurance made a difference to subscribers’ access to and utilization of health care, the challenges uninsured have in accessing and utilizing healthcare, and health care providers’ experiences of providing care to both insured and uninsured under the NHIS. Other topics included reasons for not subscribing to the NHIS, non-renewal of health insurance, recommendations for improving the NHIS, barriers to health care utilization and access. See lines 243-250 on pages 9-10 of the tracked changes version of the manuscript

Comment 7: More detail is required about how the analysis was conducted / how the themes were decided upon. E.g. How many of the authors conducted the analysis? Were the themes checked and agreed on by multiple authors?

Authors’ response: The data analysis procedure has been updated and detailed in lines 264-286 on pages 11-12 of the tracked changes version of the manuscript.

Comment 8: The themes aren’t clearly defined, with a lot of overlap between them.

The authors split the themes into factors motivating health care access and barriers to healthcare access. However, affordability of health care cost appears to be both a barrier and a facilitator and by splitting the discussion between the two themes the importance of this finding seems to get lost. Perhaps the authors may consider discussing barriers and facilitators within themes rather than having them as the main themes. E.g. create a theme “affordability of health care cost” and then discuss the barriers and facilitators of this as well as the differences / similarities between the insured and uninsured.

Authors’ response: The suggestion in this comment has been included in the revised manuscript. We have created the theme “Affordability of health care cost”. See page 15 ff.

Comment 9: In the methods the authors mention that health workers were interviewed but there is no further information about what they were interviewed about and there is no mention of this in the results?

Authors’ response: The data obtained from health workers have been included in the results as in seen in the sub-heading “Service Providers Views and Experiences of providing Care to NHIS Subscribers and Non-Subscribers”. See pages 21-23 of the tracked changes version of the manuscript.

Comment 10: Many of the issues discussed under “reasons for non-subscription and recommendations…” were discussed under other headings so this doesn’t seem like a separate theme. As mentioned above, these issues can be discussed under more specific themes.

Authors’ response: This has now been made more specific on pages 23-24 of the tracked changes version of the manuscript.

Comment 11: At the start of the discussion the authors mention views of health professionals but these weren’t actually discussed in the results.

Authors’ response: This has been rectified as is seen in the Results sub-heading “Service Providers Views and Experiences of providing Care to NHIS Subscribers and Non-Subscribers” on pages 21 23 of the tracked changes version of the manuscript. It has also been included in the discussion.

Other revisions undertaken by authors’ during this revised submission 

The manuscript has been proofread for clarity and to also address any typographical and grammatical errors.

---

## [Editor Report · Decision Letter 1]

13 Nov 2020

PONE-D-20-18400R1

The shared experiences of insured members and the uninsured in health care access and utilization under Ghana's National Health Insurance Scheme:  Evidence from the Hohoe Municipality

PLOS ONE

Dear Dr. Adam Fusheni,

Thank you for submitting your manuscript to PLOS ONE. After careful consideration, we feel that it has merit but does not fully meet PLOS ONE’s publication criteria as it currently stands. Therefore, we invite you to submit a revised version of the manuscript that addresses the points raised during the review process.

Although your revised manuscript has addressed all the concerns of the reviewers including myself, the following need to be further addressed before the manuscript will be ready for publication:

All proper nouns including names of organizations, places, titles of people, names of journals in the reference list, etc need to be capitalized. These include Sustainable Development Goals (SDGs), Minister for Employment and Social Welfare, Hohoe sub-Municipality etcPlease indicate the source of the figure used and the authorization obtained to use it in your manuscript.The figure title normally goes below the figure, not aboveAnderson uses "enabling factors" in the model so please use same consistently throughout the paper. 

We look forward to receiving your revised manuscript.

Kind regards,

Joyce Addo-Atuah, PhD

Academic Editor

PLOS ONE

---

## [Author Response · Author response to Decision Letter 1]

15 Nov 2020

Detail Revision notes on revised Manuscript

Manuscript ID: PONE-D-20-18400

Title: The shared experiences of insured members and the uninsured in health care access and utilization under Ghana's National Health Insurance Scheme: Evidence from the Hohoe Municipality

Below, we explain how we have addressed each of the comments of the reviewers.

• All proper nouns including names of organizations, places, titles of people, names of journals in the reference list, etc need to be capitalized. These include Sustainable Development Goals (SDGs), Minister for Employment and Social Welfare, Hohoe sub-Municipality etc

Authors Response: This has now been addressed throughout the manuscript in tracked changes. See the Revised Manuscript with Track Changes.

• Please indicate the source of the figure used and the authorization obtained to use it in your manuscript.

We did indicate the source of the manuscript in the figure title in the initial and first round revised versions on page 6 with reference [31]. However, due to authorization issues, we have decided to remove the figure in the revised manuscript.

• The figure title normally goes below the figure, not above

Thank you for drawing our attention but in both the initial original submission and first round revision, the title of the figure was below where the figure was to be inserted.

• Anderson uses "enabling factors" in the model so please use same consistently throughout the paper.

Thank you very much for the comment. We have now used “enabling factors” throughout the paper.

We hope we have addressed the comments in the manuscript to the satisfaction of both the Academic Editor and the reviewer(s).

Once again, thank you for giving us the opportunity to revise the manuscript.

---

## [Editor Report · Decision Letter 2]

30 Nov 2020

PONE-D-20-18400R2

The shared experiences of insured members and the uninsured in health care access and utilization under Ghana's National Health Insurance Scheme:  Evidence from the Hohoe Municipality

PLOS ONE

Dear Dr. Adam Fusheini,

Thank you for submitting your manuscript to PLOS ONE. After careful consideration, we feel that it has merit but does not fully meet PLOS ONE’s publication criteria as it currently stands. Therefore, we invite you to submit a revised version of the manuscript that addresses the points raised during the review process.

Your manuscript needs a lot of editing to meet the quality standards of this journal. Hence please pay attention to the following and all related instances in your manuscript to improve its overall editorial quality suitable for publication in this journal:

1) Spellings--eg. Minister not Minster

2) Volta Region is a proper noun so please write it as written here; kindly check all such instances of region and district names etc which should be written as proper nouns

3) Health Care Service Utilization Model is a proper noun so to be written as shown

4) Hohoe sub-municipality is the right way of writing this; please correct all instances of Hohoe-sub municipal etc

5) Please be consistent with numbers in your manuscript. Some parts of the manuscript have the numbers written in figures, other parts written out in words.

6) Low-And Middle-Income-Countries  should be written as low and middle-income countries (LMIC)

7) Also please be consistent with the formatting of your headings and sub-headings throughout the manuscript, ensuring that sub-headings at the same level, eg all first-level sub-headings or all second-level sub-headings in the manuscript have the same formatting style, eg. capitalization, use of bolding etc

8) Journal  names in the reference list are usually in italics to make them stand out.

We look forward to receiving your revised manuscript.

Kind regards,

Joyce Addo-Atuah, PhD

Academic Editor

PLOS ONE

---

## [Author Response · Author response to Decision Letter 2]

2 Dec 2020

Authors Response: The issues raised have now been addressed throughout the manuscript in tracked changes. See the Revised Manuscript with Track Changes.

• Your manuscript needs a lot of editing to meet the quality standards of this journal. Hence please pay attention to the following and all related instances in your manuscript to improve its overall editorial quality suitable for publication in this journal:

1) Spellings--eg. Minister not Minster (This has now been corrected. See line 45, page 2 of the revised manuscript with tracked changes)

2) Volta Region is a proper noun so please write it as written here; kindly check all such instances of region and district names etc which should be written as proper nouns (This has now been corrected throughout the revised manuscript and all other proper nouns written in the correct manner)

3) Health Care Service Utilization Model is a proper noun so to be written as shown (This has now been written as shown in the revised manuscript. See lines 111-113 on page 5 of the revised manuscript wit tracked changes, and also on pages 22 and 24).

4) Hohoe sub-municipality is the right way of writing this; please correct all instances of Hohoe-sub municipal etc (All instances of Hohoe-sub municipal has been corrected throughout the manuscript. See the revised tracked changed version).

5) Please be consistent with numbers in your manuscript. Some parts of the manuscript have the numbers written in figures, other parts written out in words. (This has now been revised in the manuscript. See the revised tracked change version).

6) Low-And Middle-Income-Countries should be written as low and middle-income countries (LMIC) (This has been corrected. Please, see line 4 on page 1 of the revised tracked change version).

7) Also please be consistent with the formatting of your headings and sub-headings throughout the manuscript, ensuring that sub-headings at the same level, eg all first-level sub-headings or all second-level sub-headings in the manuscript have the same formatting style, eg. capitalization, use of bolding etc (This has been changed in the revised version of the manuscript-see in tracked changes).

8) Journal names in the reference list are usually in italics to make them stand out. (All the journal names are now in Italics. See the reference list in the tracked change version).

We hope we have addressed the comments in the manuscript to the satisfaction of both the Academic Editor and the reviewer(s). The manuscripts has also been proofread.

Once again, thank you for giving us the opportunity to revise the manuscript.

---

## [Editor Report · Decision Letter 3]

4 Dec 2020

The shared experiences of insured members and the uninsured in health care access and utilization under Ghana's National Health Insurance Scheme:  Evidence from the Hohoe Municipality

PONE-D-20-18400R3

Dear Dr. Adam Fusheini,

We’re pleased to inform you that your manuscript has been judged scientifically suitable for publication and will be formally accepted for publication once it meets all outstanding technical requirements.

Kind regards,

Joyce Addo-Atuah, PhD

Guest Editor

PLOS ONE

Additional Editor Comments (optional):

Thank you for addressing all concerns raised
---

## [Editor Report · Acceptance letter]

9 Dec 2020

PONE-D-20-18400R3 

The shared experiences of insured members and the uninsured in health care access and utilization under Ghana's National Health Insurance Scheme:  Evidence from the Hohoe Municipality 

Dear Dr. Fusheini:

I'm pleased to inform you that your manuscript has been deemed suitable for publication in PLOS ONE. Congratulations! Your manuscript is now with our production department. 

Kind regards, 

on behalf of

Dr. Joyce Addo-Atuah 

Guest Editor

PLOS ONE